# GeoX: Geometric Problem Solving Through Unified Formalized Vision-Language Pre-Training

**Renqiu Xia**[1,2,*], **Mingsheng Li**[2,3,*], **Hancheng Ye**[2], **Wenjie Wu**[1], **Hongbin Zhou**[2],
**Jiakang Yuan**[2,3], **Tianshuo Peng**[2,4], **Xinyu Cai**[2], **Xiangchao Yan**[2], **Bin Wang**[2], **Conghui He**[2],
**Botian Shi**[2], **Tao Chen**[3,✉], **Junchi Yan**[1,2], **Bo Zhang**[2,‡,✉]

[1] School of Computer Science & Artificial Intelligence, Shanghai Jiao Tong University,
[2] Shanghai Artificial Intelligence Laboratory,
[3] School of Information Science and Technology, Fudan University,
[4] MMLab, The Chinese University of Hong Kong

\* Equal Contribution, ✉ Corresponding Authors, ‡ Project Leader

## Abstract

Despite their proficiency in general tasks, Multi-modal Large Language Models (MLLMs) struggle with automatic Geometry Problem Solving (GPS), which demands understanding diagrams, interpreting symbols, and performing complex reasoning. This limitation arises from their pre-training on *natural images and texts*, along with the *lack of automated verification* in the problem-solving process. Besides, current geometric specialists are limited by their *task-specific designs*, making them less effective for broader geometric problems. To this end, we present GeoX, a multi-modal large model focusing on geometric understanding and reasoning tasks. Given the significant differences between geometric diagram-symbol and natural image-text, we introduce **unimodal pre-training** to develop a diagram encoder and symbol decoder, enhancing the understanding of geometric images and corpora. Furthermore, we introduce **geometry-language alignment**, an effective pre-training paradigm that bridges the modality gap between unimodal geometric experts. We propose a **Generator-And-Sampler Transformer** (GS-Former) to generate discriminative queries and eliminate uninformative representations from unevenly distributed geometric signals. Finally, GeoX benefits from visual instruction tuning, empowering it to take geometric images and questions as input and generate verifiable solutions. Experiments show that GeoX outperforms both generalists and geometric specialists on publicly recognized benchmarks, such as GeoQA, UniGeo, Geometry3K, and PGPS9k. Our code is available at https://github.com/Alpha-Innovator/GeoX

## 1 Introduction

Large Language Models (LLMs) (Touvron et al., 2023a; Ouyang et al., 2022) and their multi-modal extensions (MLLMs) (Liu et al., 2024; Chen et al., 2024; OpenAI, 2023; Anthropic, 2024) have demonstrated exceptional abilities to effectively handle a wide range of general domain tasks, such as cross-modal retrieval (Caffagni et al., 2024; Zhang et al., 2023a; Wang et al.; Xia et al., 2024a;b), visual question answering (Wu & Xie, 2024; Huang et al., 2025; Ye et al., 2022), and summarization (Bianco et al., 2023; Rotstein et al., 2023). With the increasing focus on Artificial General Intelligence (AGI), both LLMs and MLLMs are making inroads into specialized domains such as mathematics reasoning (Imani et al., 2023; Wang et al., 2024a;b), demonstrating promising performance improvements.

Plane geometry is a pivotal and unique branch of mathematics that requires the integration of multi-modal data as well as knowledge from different scientific fields, such as theorem proving (Trinh et al., 2024) and algebraic computation (Faulstich & Oster, 2024). However, developing AI systems to automatically solve geometry problems is challenging due to the inherent complexity of both visual

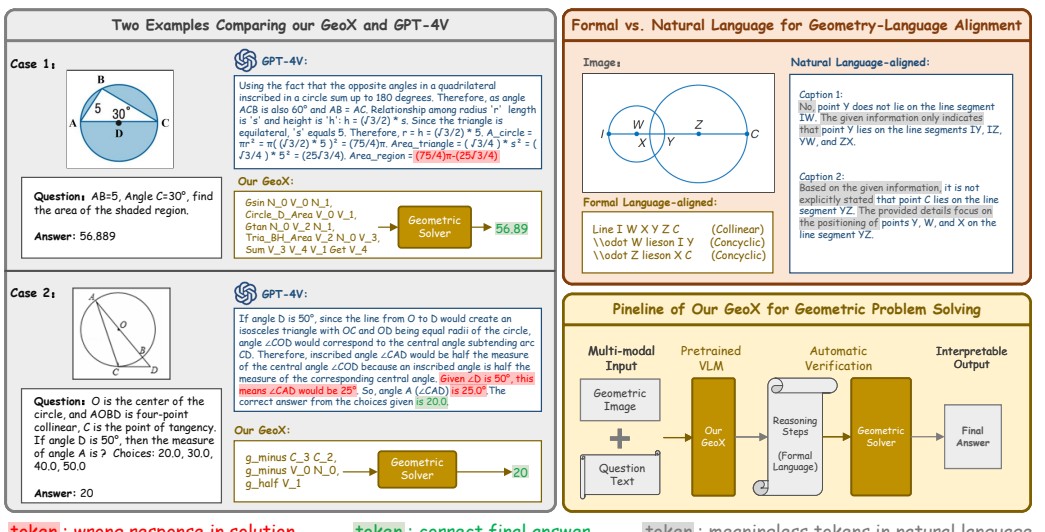

Figure 1: Highlights of GeoX: 1) Comparison between GPT-4V (OpenAI, 2023) and GeoX: GPT-4V often fails to provide the expected results or solving approaches. Besides, verifying GPT-4V's solutions is labor-intensive, requiring expert knowledge and step-by-step analysis. 2) Comparison between formal and natural (informal) language: Unlike existing works (Gao et al., 2023; Zhang et al., 2024) that use natural language, we advocate for formal language due to its effectiveness and verifiability, making it more suitable for geometric tasks. 3) GeoX solves geometric tasks in a unified format by taking geometric images and questions as input, generating verifiable program sequences, and performing solving with a solver.

and language modalities. Previous works (Peng et al., 2023; Wu et al., 2024) rely on additional detection models and make decisions based on manually crafted rules, but are often criticized for their *complexity* (Zhang et al., 2023b). On the other hand, NGS (Chen et al., 2021), Geoformer (Chen et al., 2022), and PGPSNet (Zhang et al., 2023c) focus on predicting program sequences, yet they often suffer from *poor adaptability* due to their *task-specialized model designs* and *limited ability* in modeling complex geometric diagrams and problems.

Although MLLMs (Shi et al., 2024; Lu et al.) have made significant progress in multi-modal mathematical reasoning, their performance still lags behind that of specialized geometry models. Notably, they sometimes exhibit an interesting phenomenon where they generate a correct answer accompanied by an *incorrect solution process or solving approach*, as shown in Fig. 1. Besides, we observe that using natural language to describe geometric diagrams introduces a significant amount of redundant information. In contrast, formal descriptions are more concise and clear, providing necessary information about symbols, shapes, numbers, and their relationships, making them better suited for geometric multimodal pre-training. To this end, we argue that effectively leveraging multimodal information from both visual and textual sources through formalized pre-training is meaningful in mitigating the challenges that MLLMs face when solving geometric problems.

However, combining visual and symbolic information for pre-training to boost the ability of GPS is challenging, due to the following two reasons: **1) Large Domain Gap for Geometric Understanding.** Prior works (Gao et al., 2023; Shi et al., 2024) adopt a frozen CLIP ViT (Radford et al., 2021) as the diagram encoder, which is trained on natural images rich in colors and textures. However, geometric diagrams are usually monochrome, composed of elements like lines, shapes, and symbols, exhibiting a significant domain discrepancy. **2) Uninformative Representations for Geometric Reasoning.** In geometric images, useful information is concentrated in specific areas, while other regions are uninformative and considered noise. The inability to handle this uneven distribution of geometric information leads to suboptimal performance.

To address these challenges, we propose GeoX, a geometry-centric large model that can comprehend geometric problems and solve geometry tasks in a unified formulation. To this end, we propose a formalized training scheme that consists of three progressive stages: unimodal pre-training, formalized geometry-language alignment, and visual instruction tuning. In the first stage, as introduced in Sec. 3.2, we focus on integrating a visual encoder with prior knowledge of geometry by masked auto-

encoding. At the same time, we train a geometric decoder in an auto-regressive manner to enhance its comprehension of the geometry language, which is interleaved with numbers, symbols, and words. Furthermore, solving geometric problems often requires not just recognizing shapes or symbols but also reasoning their interactions and implications. Thus, as described in Sec. 3.3, we introduce geometry-language alignment, which utilizes formalized descriptions instead of natural language captions, offering a new perspective to effectively align geometry-semantic features. We present a Generator-and-Sampler Transformer (GS-Former), capable of generating geometry content-aware queries and removing uninformative representations under the guidance of semantic learning. In Sec. 3.4, to enable GeoX to generate solutions based on the input geometric problem and image, we adopt end-to-end visual instruction tuning to obtain the ultimate model. Furthermore, in Appendix A, we theoretically explain why the proposed formalized pre-training is more effective in GPS tasks.

In Sec. 4, we conduct extensive experiments on four widely recognized benchmarks to evaluate GeoX's ability in reasoning complex and diverse geometric problems, where our approach achieves state-of-the-art results. Insightful analyses and ablation experiments are performed to further validate the effectiveness of our method.

Our contributions can be summarized as follows:

- Our study reveals the large potential of formalized visual-language pre-training in enhancing geometric problem-solving abilities. To enable the formalized pre-training, we propose GeoX, aiming to build geometric generalist models by modeling geometric tasks into a unified formulation.

- We analyze the unique challenges in the field of geometry problem solving and propose GS-Former, which effectively bridges the modality gap between geometric diagrams and formalized language.

- Compared with previous generalist and specialized models, our GeoX achieves competitive performance on GeoQA, UniGeo, Geometry3K, and PGPS9K, further demonstrating GeoX as a strong baseline model for solving geometric problems and motivating future research.

## 2 RELATED WORKS

**Multi-modal Large Language Models.** The past year has witnessed the notable success of Large Language Models (LLMs) families (Ouyang et al., 2022; Touvron et al., 2023a;b; Team, 2023), showcasing near-human performance across diverse tasks. Concurrently, researchers have made significant efforts to extend the abilities of LLMs in handling visual-related tasks, contributing to the flourishing of Multimodal Large Language Models (MLLMs) (Bai et al., 2023; Achiam et al., 2023; Reid et al., 2024). MLLMs typically adopt a cross-modal projector as the bridge to reconcile the modality gap between visual encoder and LLM, such as Q-former (Li et al., 2023b) or linear layers (Liu et al., 2024). Although MLLMs have demonstrated impressive performance in conventional vision-language tasks (Han et al., 2024; Xia et al., 2023; Li et al., 2023c), they yield unsatisfactory results when addressing multimodal mathematical problems involving geometric diagrams and symbols. Besides, G-LLaVA (Gao et al., 2023) and MAVIS (Zhang et al., 2024) train LLM on the constructed geometry datasets with descriptions in natural language form. Recently, Chimera (Peng et al., 2024) proposes using the general-expert collaboration masking method to effectively integrate expert knowledge into a general MLLMs. However, as illustrated in Fig. 1, these works face two issues: *1) unable to provide the answer as required*, and *2) incorrect solving steps that still result in correct answers*. Furthermore, verifying the solving process of MLLMs is extremely costly since it requires human experts from geometric knowledge and a step-by-step examination. To this end, we propose GeoX, which solves geometric tasks in a unified formulation and predicts verifiable solutions.

**Geometry Problem Solving (GPS)** is a long-standing yet challenging task in mathematics, requiring models with the ability to understand geometric elements and reason with logic. Existing automatic systems for GPS fall into two categories: rule-based approaches and neural approaches. Rule-based approaches (Seo et al., 2015; Sachan & Xing, 2017; Lu et al., 2021; Peng et al., 2023; Wu et al., 2024) rely on external tools like OCR to parse diagrams into texts, which are then used for logical reasoning based on path search and condition matching. Although these methods have shown satisfactory performance in GPS, they are *heavily dependent on manually crafted rules*, making them difficult

to generalize to diverse geometry scenarios. Neural approaches use networks to predict solving steps via program sequences, which are then executed by the solver. For example, NGS (Chen et al., 2021) and Geoformer (Chen et al., 2022) introduce auxiliary self-supervised tasks to refine diagram representations, with experiments on GeoQA (Chen et al., 2021) and UniGeo (Chen et al., 2022) demonstrating the effectiveness of their methods. Other methods, such as PGPSNet (Zhang et al., 2023c) and LANS (Zhang et al., 2023b), integrate structural and semantic clauses into solving process and utilize specially designed decoders to achieve better performance on both Geometry3K (Lu et al., 2021) and PGPS9K (Zhang et al., 2023c). While these geometry specialists have shown impressive performance, their *uniquely designed models* for specialized datasets limit their ability to solve broader geometric tasks. In contrast, we introduce the unified formalized vision-language pre-training for general geometric tasks, achieving superior results across diverse benchmarks compared to previous methods on GPS.

## 3 FORMALIZED VISION-LANGUAGE PRE-TRAINING

### 3.1 METHOD OVERVIEW

To tackle complicated plane geometry problems, we introduce GeoX, adopting a formalized pre-training scheme consisting of three progressive stages, as illustrated in Fig. 2.

**Unimodal Pre-training.** Vanilla generalist models (OpenAI, 2023; Anthropic, 2024; Team et al., 2023; Bai et al., 2023; Peng et al., 2024; Chen et al., 2024) have poor representation capacity in the geometric domain, due to the significant gaps between non-formalized data (*e.g.*, informal text descriptions and natural images) and formalized data (*e.g.*, formal geometric symbols and scientific images). As a result, we propose unimodal pre-training in Sec. 3.2, aiming to enhance the GeoX's ability to understand geometric diagrams and symbols.

**Geometry-Language Alignment.** To facilitate the aforementioned pre-trained unimodal models for performing cross-modal alignment, we propose an effective Generator-and-Sampler Transformer (GS-Former), which is trained using pairs of geometric diagrams and formal language descriptions, as detailed in Sec. 3.3.

**End-to-end Instruction Tuning.** After geometry-language alignment, the ultimate model is required to generate solutions based on the given geometric problems and images. To this end, we tune GeoX in an end-to-end visual instruction tuning manner (as introduced in Sec. 3.4), boosting its capacity to comprehend geometric problems and generate formal solution programs.

During the inference phase, the solution generated by GeoX is fed into the symbolic solver (Chen et al., 2021; Zhang et al., 2023c), which performs step-by-step operations to predict the final answer.

### 3.2 UNIMODAL PRE-TRAINING

**Geometry Encoder.** To mitigate the deficiencies of the existing visual encoders in comprehending geometric images, we collect more than 120K diagrams[1] from the web and electronic textbooks to equip ViT with prior knowledge of geometry, abbreviated as Geo-ViT. Similar to He et al. (2022), we tune the vision encoder-decoder using the masked auto-encoding scheme, where some patches are masked and the remaining subset is fed into the visual encoder, with the original image subsequently reconstructed by a lightweight decoder. In the next stages, we only utilize the pre-trained encoder to represent geometric diagrams.

**Symbol Decoder.** Considering the capability of LLMs to follow users' instructions and handle different tasks, we utilize the decoder-only LLM as our symbol decoder to generate solutions. However, LLMs (Brown, 2020; Touvron et al., 2023b) are typically trained on general text, which lacks the specialized learning for geometry. To this end, we build a 100M-token geometric corpus based on the existing datasets (Chen et al., 2021; Lu et al., 2021; Gao et al., 2023; Zhang et al., 2023c; Chen et al., 2022; Cao & Xiao, 2022), containing a wide range of geometric problems, symbols, theorems, and so on. More details can be found in Appendix E. We choose LLEMMA-7B (Azerbayev et al., 2023) as the base model, an open-source language model for mathematics pre-trained on Proof-Pile-2 (Azerbayev et al., 2023), and further fine-tune it on the geometric corpus using a standard auto-regressive language modeling objective, resulting in Geo-LLM-7B.

---

[1] https://huggingface.co/datasets/U4R/GeoX-data

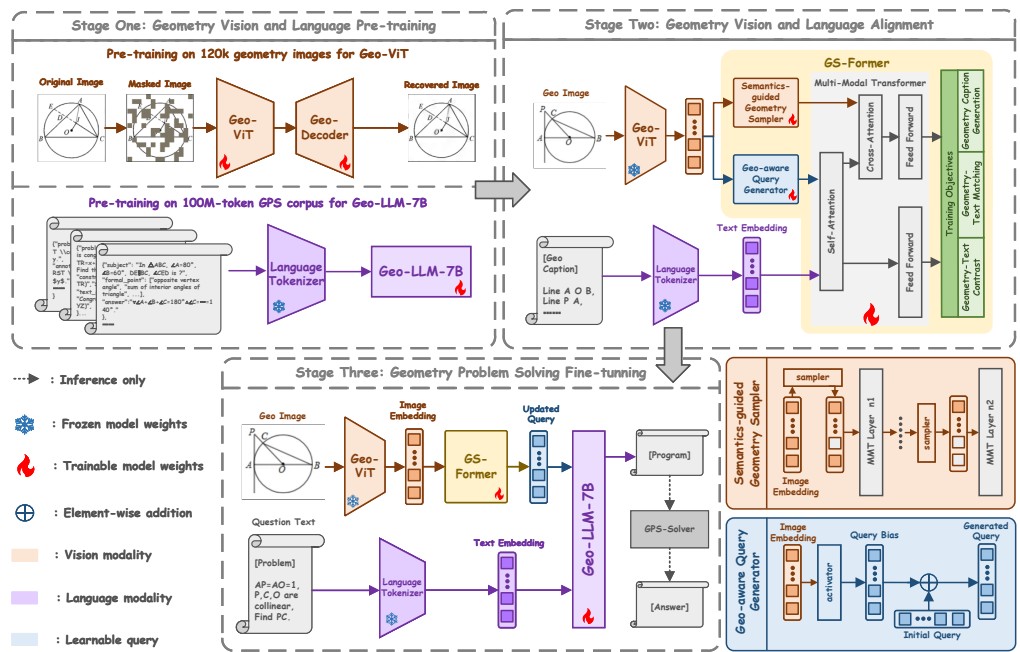

Figure 2: Overview of GeoX for training. We present a versatile method for automatic geometric problem solving through unified formalized vision-language pre-training, which comprises three progressive stages.

## 3.3 GEOMETRY-LANGUAGE ALIGNMENT

### 3.3.1 DATA ENGINE

While recent datasets (Gao et al., 2023; Zhang et al., 2024) have made strides in captioning geometric images using natural language, they often result in redundant information, as depicted in Fig. 1. In contrast, our approach emphasizes the use of formal descriptions to encapsulate the spatial structural information within geometric images. This information is implicitly represented, not explicitly stated in the problem text. Our curated dataset[1] focuses on capturing the essence of geometric imagery by detailing the relationships between the most fundamental elements (points) without explicitly annotating higher-level constructs such as squares or triangles, which can either be inferred from the relationships we describe or are directly provided in the problem text.

Our formalized diagram-caption dataset delves into the spatial relationships at a granular level, starting with the basic building blocks of geometric images. We identify and describe the relative positions and connections between points, ensuring that the spatial relationships are accurately represented. These relationships are categorized into two primary types: **1)** Collinear Relationship (*e.g.*, `line A B C` signifies that points A, B, and C are on the same line) and **2)** Concyclic Relationship (*e.g.*, `\\odot O lieson A B C` denotes that points A, B, and C are on the same circle with center O). The dataset encompasses 6232 geometric images sourced from the internet, meticulously annotated by a team of 10 experts over a period of 200 hours. Moreover, we provide concrete examples along with comprehensive explanations of formalized diagram-caption pairs in Appendix C.

### 3.3.2 GENERATOR-AND-SAMPLER TRANSFORMER

With the formalized geometry-language dataset, GeoX learns a unified representation space for geometry and formalized language through the Generator-and-Sampler Transformer (GS-Former), which includes a Geo-aware Query Generator (GQG) and a Multi-Modal Transformer.

**Geo-aware Query Generator.** Both Resampler (Alayrac et al., 2022; Li et al., 2023a) and Q-Former (Li et al., 2023b; Dai et al., 2023) extract visual features using a set of static query tokens, which are randomly initialized and regarded as model parameters. However, these queries, which remain the same for different diagrams, fail to capture discriminative features unique to individual samples. Thus, we introduce the Geo-aware Query Generator (GQG), which incorporates contextual information to dynamically generate queries.

To be specific, GQG utilizes visual features from the encoder and aggregates contextual information through an attention-based module and pooling operation. The contextual features then are projected and added with learnable queries (Li et al., 2023b), which builds a connection between the learnable queries and the geometric content. Our empirical results demonstrate the effectiveness of GQG, resulting in improved performance.

**Multi-Modal Transformer** comprises $N_L$ layers, each containing a self-attention block, a cross-attention block, and a feed-forward network. Queries within each layer initially interact with paired formal captions and are then fed into the cross-attention block to extract visual features.

To handle the uneven information distribution in geometric images as described in Sec. 1, we introduce the Semantics-guided Geometry Sampler (SGS), which dynamically removes uninformative visual representations guided by vision-language alignment. Specifically, SGS is tasked with predicting a binary mask $M = \{m_j^i \mid i \in K, j \in N\}$, with each $m_j^i \in \{0, 1\}$ determining whether to retain or discard visual representations. Here, $K$ represents the layer number and $N$ denotes the number of patches. This module receives the previous mask $M^{i-1}$ and visual features as inputs, using a linear layer to obtain retention probabilities $P^i$. To enable differentiable sampling from probabilities, we use the reparameterization (Jang et al., 2016) with Gumbel-Softmax:

$$M^i = M^{i-1} \odot \text{Gumbel-Softmax}(P^i), \tag{1}$$

where $\odot$ is the Hadamard product, $i$ and $i - 1$ represents the previous stage and current stage. A notable feature of our GS-Former is its capability to progressively drop noisy and semantically irrelevant features under the guidance of geometric language alignment. This is achieved by initially setting all elements of the decision mask to 1, followed by inserting the SGS block at subsequent layers. Additionally, GS-Former is initialized with weights from pre-trained BERT models (Kenton & Toutanova, 2019), except for the SGS and cross-attention layers, which are initialized randomly.

Inspired by BLIP-2 (Li et al., 2023b), we introduce a multimodal alignment loss $\mathcal{L}_{align}$ to optimize GS-Former, incorporating three training objectives: Geometry-Text Contrast and Geometry-Text Matching, both designed to align features between geometric diagrams and formal text, along with Geometry Caption Generation, aimed at generating formal captions based on visual information. We further impose a sparsification term $\mathcal{L}_{spr}$ into the overall optimization objective to prevent trivial solutions where all mask values $m_j^i$ are set to 1:

$$\mathcal{L}_p = \mathcal{L}_{align} + \lambda \mathcal{L}_{spr}, \text{ where } \mathcal{L}_{spr} = \frac{1}{KN} \sum_{i \in K, j \in N} \left\| m_j^i \right\|_1 . \tag{2}$$

## 3.4 END-TO-END VISUAL INSTRUCTION TUNING

To enable the model to handle geometry-centric tasks, we continue the training with end-to-end visual instruction tuning, directing the ultimate model to generate solutions. As illustrated in Fig. 2, we feed the diagrams into the pre-trained Geo-ViT together with GS-Former, to obtain the semantically aligned geometry features $F_g$. Besides, we utilize a trainable projection head $\mathcal{W}$ to project $F_g$ into the language embedding space and obtain visual tokens $T_g$. Geo-LLM, serving as a decoder for various geometry tasks, takes both visual tokens $T_g$ and instruction tokens $T_p$ as input, and generates solutions in an auto-regressive manner. Our training objective is to optimize the GeoX so that the likelihood of the target sequence $S = \{s_{i,i \in [1:L]}\}$ is maximized given the visual input $T_g$ and instruction $T_p$.

In practice, GeoX is trained using cross-entropy loss $\mathcal{L}_t$ defined as follows, which optimizes the model to predict the $l$-th token $s_l$ given preceding token sequences $s_{i,i \in [1:l-1]}$:

$$\mathcal{L}_t = -\sum_l \log P(s_l | s_{i,i \in [1:l-1]}; T_g; T_p). \tag{3}$$

## 4 EXPERIMENTS

### 4.1 DATASETS, METRICS, AND IMPLEMENTATION DETAILS

**Datasets.** To assess the effectiveness of GeoX, we conduct experiments on four widely recognized geometry benchmarks: GeoQA (Chen et al., 2021), UniGeo (Chen et al., 2022), Geometry3K (Lu

Table 1: Comparison of various methods on the GeoQA benchmark with different accuracy metrics.

| Methods | Metric | Total | Angle | Length |
|---|---|---|---|---|
| *Generalists* | | | | |
| mPLUG-Owl2 (Ye et al., 2023) | | 16.0 | 16.5 | 15.9 |
| LLaVA-v1.5 (Liu et al., 2024) | | 20.7 | 20.9 | 19.8 |
| Qwen-VL (Bai et al., 2023) | | 24.4 | 23.7 | 24.4 |
| GPT-4V (OpenAI, 2023) | | 43.4 | 39.3 | 49.8 |
| *Specialists* | Top-1 | | | |
| LLaVA-v1.5 (Liu et al., 2024)+Solver | | 9.4 | 14.9 | 3.2 |
| NGS(Chen et al., 2021) | | 46.3 | - | - |
| UniMath-T5(Liang et al., 2023) | | 49.6 | - | - |
| UniMath-Flan-T5(Liang et al., 2023) | | 50.0 | - | - |
| GeoX (Ours) | | **54.9** | **62.8** | **45.2** |

| Methods | Metric | Total | Angle | Length |
|---|---|---|---|---|
| *Specialists* | | | | |
| LLaVA-v1.5 (Liu et al., 2024)+Solver | | 29.2 | 40.5 | 15.9 |
| FiLM(Perez et al., 2018) | | 31.7 | 34.0 | 29.7 |
| RN(Santoro et al., 2017) | | 38.0 | 42.8 | 32.5 |
| MCAN(Yu et al., 2019) | | 39.7 | 45.0 | 34.6 |
| BERT (Kenton & Toutanova, 2019) | Top-10 | 54.7 | 65.8 | 42.1 |
| NGS(Chen et al., 2021) | | 56.9 | 69.8 | 39.2 |
| Geoformer(Chen et al., 2022) | | 60.3 | 71.5 | 49.1 |
| DPE-NGS(Cao & Xiao, 2022) | | 62.7 | 74.9 | 47.7 |
| SCA-GPS(Ning et al., 2023) | | 64.1 | 74.9 | 50.1 |
| GeoX (Ours) | | **69.0** | **78.2** | **58.0** |

Table 2: Comparison of model performance on UniGeo for geometry calculation and proof problems.

| Methods | Metric | Calculation(%) | | | Proving (%) | | | | | |
|---|---|---|---|---|---|---|---|---|---|---|
| | | All ↑ | Angle ↑ | Length ↑ | All ↑ | Par. ↑ | Tri. ↑ | Qua. ↑ | Con. ↑ | Sim. ↑ |
| *Generalists* | | | | | | | | | | |
| mPLUG-Owl2 (Ye et al., 2023) | | 18.7 | 18.7 | 19.1 | - | - | - | - | - | - |
| LLaVA-v1.5 (Liu et al., 2024) | | 24.0 | 26.4 | 21.6 | - | - | - | - | - | - |
| Qwen-VL (Bai et al., 2023) | | 24.4 | 24.2 | 25.4 | - | - | - | - | - | - |
| GPT-4V (OpenAI, 2023) | | 47.9 | 45.8 | 51.6 | - | - | - | - | - | - |
| *Specialists* | Top-1 | | | | | | | | | |
| LLaVA-v1.5 (Liu et al., 2024)+Solver | | 16.1 | 19.2 | 13.1 | 1.0 | 0.0 | 1.1 | 0.4 | 0.2 | 3.0 |
| Geoformer (Chen et al., 2022) | | 46.8 | 57.8 | 35.0 | 51.3 | 13.9 | 63.8 | 20.4 | 56.1 | 64.0 |
| UniMath-T5-base (Liang et al., 2023) | | - | - | - | 82.9 | - | - | - | - | - |
| UniMath-Flan-T5-base (Liang et al., 2023) | | - | - | - | 83.0 | - | - | - | - | - |
| GeoX (Ours) | | **54.4** | **63.1** | **43.1** | **97.8** | **77.8** | **100.0** | **95.4** | **99.5** | **99.2** |
| *Specialists* | | | | | | | | | | |
| LLaVA-v1.5 (Liu et al., 2024)+Solver | | 43.0 | 51.3 | 35.3 | 11.3 | 0.0 | 16.2 | 5.0 | 2.9 | 27.5 |
| BERT (Kenton & Toutanova, 2019) | | 52.0 | 63.1 | 39.2 | 48.1 | 15.4 | 48.0 | 31.7 | 49.5 | 75.1 |
| NGS (Chen et al., 2021) | Top-10 | 51.9 | 63.6 | 38.8 | 47.4 | 11.2 | 46.9 | 31.3 | 48.3 | 77.6 |
| Geoformer (Chen et al., 2022) | | 62.5 | 75.5 | 48.8 | 56.4 | 19.4 | 69.4 | 20.4 | 60.3 | 75.0 |
| GeoX (Ours) | | **68.6** | **76.7** | **58.3** | **99.5** | **97.2** | **100.0** | **97.7** | **100.0** | **100.0** |

et al., 2021), and PGPS9K (Zhang et al., 2023c). GeoQA comprises 4,998 geometry problems sourced from Chinese middle school exams, including different types of problems, such as angles, lengths, and areas. Following Liang et al. (2023); Gao et al. (2023), we use the English version to maintain linguistic consistency with other datasets. UniGeo features 4,998 calculation problems from GeoQA and 9,543 proving problems from high school textbooks and online resources, providing a comprehensive benchmark for evaluating geometry reasoning abilities. Both Geometry3K and PGPS9K include high-quality diagrams and detailed annotations.

**Metrics.** We adopt the same evaluation metrics used in previous studies to ensure fair comparability. Following Chen et al. (2021) and Chen et al. (2022), we assess the model's performance on GeoQA and UniGeo with top-1 and top-10 accuracies. For evaluation on Geometry3K and PGPS9K, we apply three metrics to assess the performance of GeoX: Completion, Choice, and Top-3, as introduced in Zhang et al. (2023c). To evaluate MLLMs in solving complex geometry problems, such as Qwen-VL (Bai et al., 2023) and GPT-4V (OpenAI, 2023), we follow LANS (Zhang et al., 2023b) by utilizing Completion (which requires models to provide answers directly) and Choice (which involves selecting from given options).

**Implementation Details.** We optimize the diagram encoder using MAE VIT-B (He et al., 2022) checkpoints, training it for 800 epochs with a batch size of 256 and an initial learning rate of 6.4e-5. We initialize the symbol decoder with LLEMMA-7B (Azerbayev et al., 2023) weights and train it for 5 epochs with a batch size of 32 and an initial learning rate of 1e-6. For geometry-language alignment, we train the GS-Former for 360 epochs with a batch size of 256 and an initial learning rate of 1e-4. The number of queries in GS-Former is set to 8. Additional details regarding visual instruction tuning can be found in Appendix F. We implement GeoX using PyTorch and conduct experiments on more than eight A100 (80GB) GPUs. During inference, we employ a beam search size of 10, consistent with Zhang et al. (2023c) and Chen et al. (2021).

## 4.2 COMPARISONS WITH STATE-OF-THE-ART METHODS

**Performance Comparison with Generalist Models.** As to multimodal large models, LLaVA-v1.5 (Liu et al., 2024), mPLUG-Owl2 (Ye et al., 2023), Qwen-VL (Bai et al., 2023), and GPT-4V (OpenAI, 2023) exhibit strong cross-modal reasoning abilities for general tasks. However, when applied to solve geometry tasks, these models are insufficient. Our GeoX significantly outperforms these generalists on various geometry datasets, including GeoQA (Chen et al., 2021), UniGeo (Chen et al., 2022), Geometry3K (Lu et al., 2021), and PGPS9K (Zhang et al., 2023c). As indicated in

Table 3: Performance comparison on Geometry3K and PGPS9K.

| Methods | Geometry3K | | | PGPS9K | | |
|---|---|---|---|---|---|---|
| | $Completion \uparrow$ | $Choice \uparrow$ | $Top-3 \uparrow$ | $Completion \uparrow$ | $Choice \uparrow$ | $Top-3 \uparrow$ |
| *Generalists* | | | | | | |
| mPLUG-Owl2 (Ye et al., 2023) | 2.2 | 26.7 | - | 3.0 | 26.4 | - |
| LLaVA-v1.5 (Liu et al., 2024) | 2.9 | 22.9 | - | 1.8 | 21.8 | - |
| Qwen-VL (Bai et al., 2023) | 2.5 | 27.5 | - | 1.4 | 24.7 | - |
| GPT-4V (OpenAI, 2023) | 34.8 | 58.6 | - | 33.3 | 51.0 | - |
| *Specialists* | | | | | | |
| LLaVA-v1.5 (Liu et al., 2024)+**Solver** | 19.7 | 47.4 | 31.6 | 21.6 | 38.1 | 35.3 |
| GeoDRL (Peng et al., 2023) | - | 68.4 | - | - | - | - |
| NGS (Chen et al., 2021) | 35.3 | 58.8 | 62.0 | 34.1 | 46.1 | 60.9 |
| Geoformer (Chen et al., 2022) | 36.8 | 59.3 | 62.5 | 35.6 | 47.3 | 62.3 |
| InterGPS (Lu et al., 2021) | 44.6 | 56.9 | - | - | - | - |
| PGPSNet (Zhang et al., 2023c) | 48.1 | 70.1 | 65.7 | 44.4 | 57.6 | 64.8 |
| GeoX (Ours) | **58.6** | **72.5** | **69.4** | **52.7** | **63.3** | **65.4** |

Tab. 1 and Tab. 2, GeoX achieves top-1 accuracies of 54.9% and 54.4%, respectively, significantly outperforming the best generalist models. Similarly, on Geometry3K and PGPS9K in Tab. 3, GeoX achieves 58.6% and 52.7% in Completion, respectively. In comparison, GPT-4V (OpenAI, 2023) achieves 34.8% and 33.3%, while other models such as Qwen-VL (Bai et al., 2023) and LLaVA (Liu et al., 2024) perform worse.

**Performance Comparison with Specialist Models.** Compared with geometry specialists such as NGS (Chen et al., 2021), UniMath-T5 (Liang et al., 2023), Geoformer (Chen et al., 2022), DPE-NGS (Cao & Xiao, 2022), and SCA-GPS (Ning et al., 2023), GeoX demonstrates superior performance across GeoQA and UniGeo. Specifically, GeoX surpasses the best geometry specialist by +4.9% and +7.6% on GeoQA and UniGeo-Calculation, respectively. Additionally, our model achieves significant improvements over previous methods on UniGeo-proving by +14.8% and +43.1% in Tab. 2. As reported in Tab. 3, our method outperforms SOTA models on Geometry3K and PGPS9K. Notably, previous works (Zhang et al., 2023c;b) require additional image annotations (Diagram GT) as input, which is labor-consuming and contrary to experimental settings. To make a fair comparison, we remove Diagram GT and replicate these methods under the original conditions. Particularly, we fine-tune LLaVA (Liu et al., 2024) with formal language and adopt solvers for problem-solving, consistent with the approach used in GeoX. Extensive results in Tabs. 1 to 3 demonstrate the effectiveness of GeoX, achieving state-of-the-art performance across diverse scenarios.

Besides, it should be noted that G-LLaVA-7B (Gao et al., 2023) and MAVIS (Zhang et al., 2024) achieve 64.2% and 66.7% accuracy on GeoQA. However, these models can produce correct results despite errors in the solving process. In contrast, our method treats any process errors as incorrect results. To this end, we introduce a comparable metric, with detailed results provided in Appendix D.

## 4.3 QUANTITATIVE EVALUATION ON THE GPS TASK OF MATHVISTA

We provide a quantitative comparison with the model that performed best on the GPS task in MathVista (Lu et al.). To this end, we extract the Geometry subset from MathVista, referred to as MathVista-GEO. We assess these methods using the same evaluation script as MathVista, along with the evaluation strategy introduced in Appendix D. As reported in Tab. 4, GeoX is more effective in solving geometry tasks.

Table 4: Accuracy scores on *testmini* of MathVista-GEO.

| Methods | Accuracy |
|---|---|
| GPT-4V (OpenAI, 2023) | 54.8 |
| GPT-4o (OpenAI, 2024) | 66.1 |
| GeoX (Ours) | **72.6** |

## 4.4 INSIGHTFUL ANALYSES

**Effectiveness of Uni-modal Pre-training.** We compare Geo-ViT with CLIP-ViT (Radford et al., 2021), which has been widely used for GPS in previous studies (Gao et al., 2023). Additionally, we evaluate the performance of different language models in solving geometric problems, including LLAMA-2-7B, LLEMMA-7B, and our Geo-LLM-7B. As reported in Fig. 3, compared to general-purpose models or the mathematical model, our pre-trained model demonstrates superior results across various geometry benchmarks.

**Effectiveness of Geometry-Language Alignment.** As illustrated in Tab. 5, without multi-modal feature alignment, the baseline model perform poorly, achieving only 33.1% Completion on Geometry3K. The introduction of GS-Former significantly boosts performance. Moreover, our results reveal

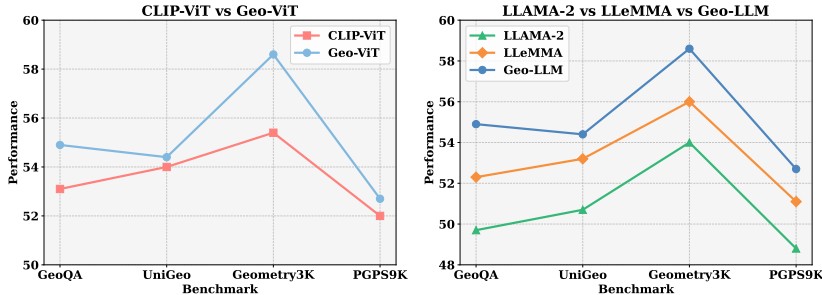

Figure 3: Effectiveness of Uni-modal Pre-training. We compare the widely used CLIP-ViT-B and our Geo-ViT-B, along with three LLM models: LLAMA-2-7B, LLEMMA-7B, and our Geo-LLM-7B.

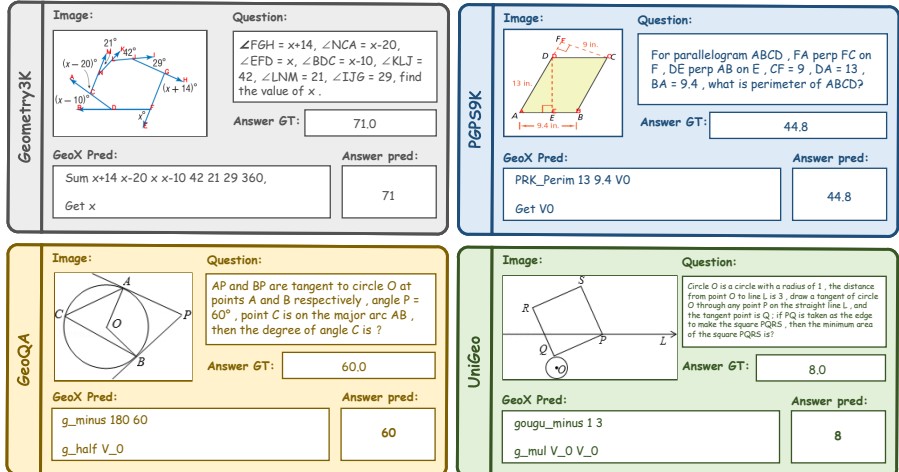

Figure 4: Visualization results on four datasets by our GeoX.

that formal language is more effective for GPS than natural language, with +2.9% improvement in Completion on Geometry3K.

Table 5: Effectiveness of geometry-language alignment.

| Module | Alignment | Language | Geometry3K | | | PGPS9K | | |
|---|---|---|---|---|---|---|---|---|
| | | | $Completion \uparrow$ | $Choice \uparrow$ | $Top-3 \uparrow$ | $Completion \uparrow$ | $Choice \uparrow$ | $Top-3 \uparrow$ |
| - | × | - | 33.1 | 54.0 | 48.2 | 31.5 | 43.6 | 50.1 |
| | × | - | 48.6 | 65.7 | 63.2 | 42.7 | 54.3 | 56.8 |
| GS-Former | ✓ | Natural | 55.7 | 71.5 | 67.2 | 52.2 | 62.2 | **67.1** |
| | ✓ | Formal | **58.6** | **72.5** | **69.4** | **52.7** | **63.3** | 65.4 |

**Ablation of Modules in GS-Former.** The results in Tab. 6 demonstrate the effectiveness of the Geo-aware Query Generator (GQG) and Semantics-guided Geometry Sampler (SGS) within GS-Former. Adding the GQG improves Completion by +2.4% and +1.0%, while combining both designs yields the best performance. The quantitative results in Appendix B further demonstrate GS-Former's effectiveness in capturing valuable information from geometry diagrams, such as lines and symbols.

## 4.5 CASE STUDY

As shown in Fig. 4, we conduct a case study to analyze the capabilities of GeoX. GeoX tries to predict formalized program sequences composed of mathematical variables, constants, and operations, such as summation (`Sum`), subtraction (`g_minus`), perimeter calculation (`PRK_Perim`), the Pythagorean theorem (`gougu_minus`), *etc.*, which can be compiled and solved by the GPS-solver.

Furthermore, we have conducted the generalization validation of GeoX in a broader scope, including its application to geometric problem-solving from natural images. Our GeoX has demonstrated promising performance in these scenarios, indicating the potential for its generalization to even wider fields. We present some visualized examples in Fig. 5.

Table 6: Ablation study of modules in GS-Former, assessing the contribution of GQG and SGS modules when GS-Former is utilized for geometry-formal language alignment.

| Geo-aware Query Generator | Semantics-guided Geometry Sampler | Geometry3K | | | PGPS9K | | |
|:---:|:---:|:---:|:---:|:---:|:---:|:---:|:---:|
| | | $Completion \uparrow$ | $Choice \uparrow$ | $Top-3 \uparrow$ | $Completion \uparrow$ | $Choice \uparrow$ | $Top-3 \uparrow$ |
| ✗ | ✗ | 55.0 | 70.3 | 68.3 | 49.8 | 59.9 | 64.6 |
| ✓ | ✗ | 57.4 | 71.7 | 68.1 | 50.8 | 62.0 | 64.3 |
| ✓ | ✓ | **58.6** | **72.5** | **69.4** | **52.7** | **63.3** | **65.4** |

**Birthday Hat**

Image:

Question: For the birthday hat made by Xiao Lan with colored paper, if the base radius is 5 cm and the slant height is 10 cm, the lateral surface area of the hat is?

Answer GT: $50\pi$

GeoX Pred: cal_cone N_0 N_1

Answer pred: 157.08

**Revolving Door**

Image:

Question: The interior of the revolving door of a hotel entrance is composed of three glass partitions with a width of 2 meters and a height of 3 meters. The three glass partitions are placed at the same angle. If the distance between the two columns at the entrance is 2 meters, then the distance from the midpoint of the bottom of the two columns to the bottom of the central shaft is ?

Answer GT: $\sqrt{3}$

GeoX Pred: g_half N_0

gougu_minus N_2 V_0

Answer pred: 1.73

**Foldable Table**

Image:

Question: A foldable square table is shown in The figure. Given that AO=BO=50cm, CO=DO=30cm, the table is now laid flat. To make the tabletop 40cm high from the ground, the angle between the two legs should be ?

Answer GT: 120

GeoX Pred: g_minus C_3 C_0

g_minus V_0 C_0

Answer pred: 120

**Bicycle**

Image:

Question: The figure shows a real picture and a schematic diagram of a bicycle. AB is parallel to the ground, points A, B, and D are collinear, points D, F, and G are collinear, and the seat C can be adjusted along the ray BE. It is known that ∠ABE=70°, ∠EAB=45°, the wheel radius is 30cm, and BE=40cm. Xiao Ming felt that it was more comfortable to ride when the seat C was 0.9m above the ground. At this time, the length of CE is ?

Answer GT: 24

GeoX Pred: g_double N_2
g_divide V_0 N_6
g_minus V_1 N_3

Answer pred: 24

Figure 5: Four visualized examples of geometric problem in natural images solved by our GeoX.

## 5 CONCLUSION, LIMITATIONS, AND FUTURE WORK

In this paper, we have proposed GeoX, a novel multi-modal large model specifically designed for automatic Geometry Problem Solving (GPS) tasks. GeoX verifies that formalized vision-language learning is beneficial to learn informative representations for automatic GPS tasks. GeoX can produce formalized process descriptions, which enhance the interpretability of GPS and the correctness of the solution process. Besides, extensive experimental analyses demonstrate GeoX's general capabilities on multiple geometric datasets.

## ACKNOWLEDGEMENT

The research was supported by Shanghai Artificial Intelligence Laboratory, the National Key R&D Program of China (Grant No. 2022ZD0160104), the Science and Technology Commission of Shanghai Municipality (Grant No. 22DZ1100102), the National Natural Science Foundation of China (Grant No. 92370201 and 62222607), and Shanghai Rising Star Program (Grant No. 23QD1401000).

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

# APPENDIX

The appendix mainly includes the following aspects:

- Sec. A: Theoretical analysis on the proposed formalized vision-language pre-training.
- Sec. B: More visualization results.
- Sec. C: Examples of formalized diagram-caption pairs.
- Sec. D: Additional quantitative evaluations.
- Sec. E: Data acquisition for geometric corpus.
- Sec. F: Implementation details.
- Sec. G: Further discussions and analyses.

**Codes** are released at https://github.com/Alpha-Innovator/GeoX, including the process of the training and evaluation of GeoX.

## A THEORETICAL ANALYSIS

In this section, we theoretically explain why the proposed formalized pre-training benefits more than informal pre-training methods in downstream tasks of geometry problems. First, we consider the sufficient representations for the pre-training of the Geometric Problem-Solving (GPS) models, containing the information shared between different modalities of geometry data. The definition of sufficient representations is borrowed and extended from the idea in Wang et al. (2022), We denote $T_f$ as the target formalized descriptions of samples in the pre-training dataset, while $T_{inf}$ is denoted as the informal descriptions of samples in the pre-training dataset. The representations learned from $T_f$ is denoted as $z_f$, while the representations learned from $T_{inf}$ is denoted as $z_{inf}$. The downstream task label is denoted as $T$, which is a formalized textual sequence that will be fed into the GPS-Solver for verifiable numerical solutions.

**Definition 1.** *(Sufficient Representations) The representations $z_{1,suf}$ of $y_1$ is sufficient for another task $y_2$ **if and only if** $I(z_{1,suf}, y_2) = I(y_1, y_2)$, where $z_{1,suf}$ is learned from $y_1$, and $y_1$, $y_2$ are the labels of two different prediction tasks that contain the shared information. $I(\cdot, \cdot)$ refers to the mutual information between the two variables.*

**Definition 2.** *(Minimal Sufficient Representations) The representations $z_{1,min}$ is minimal sufficient **if and only if** $I(z_{1,min}, y_2) = \min_{z_{1,suf}} I(z_{1,suf}, y_2)$.*

**Lemma 1.** $z_f$ provides more information about the downstream task $T$ than $z_{inf}$. That is, $I(z_f, T) \geq I(z_{inf}, T)$.

*Proof.* Since both $T_f$ and $T_{inf}$ are supervised learning tasks, their learned representations $z_f$ and $z_{inf}$ are both sufficient representations. However, since $T_{inf}$ only contains the semantic information without structural context that is required by the downstream tasks. Therefore, it holds that $I(z_{inf}, T) \leq I(z_{suf}, T), \forall z_f$ that is sufficient. That is, $z_{inf}$ is a minimal sufficient representation. As for $z_f$, it learns information from the formalized description and thus is more related to the downstream tasks. Consequently, we have the relationship between $z_{inf}$ and $z_f$ as follows,

$$I(z_f, T) = I(z_{inf}, T) + [I(T_f, T|z_{inf}) - I(T_f, T|z_f)]$$
$$\geq I(z_{inf}, T). \tag{4}$$

The first equation indicates that the mutual information $I(z_f, T)$ can be decomposed into the minimal mutual information $I(z_{inf}, T)$ and the information gap between $I(T_f, T|z_{inf})$ and $I(T_f, T|z_f)$, where $I(T_f, T|z_{inf})$ refers to the information about $T$ that can be observed from $T_f$ on condition of $z_{inf}$. Since $T_f$ contains more formalized information related to $T$ and $I(z_{inf}, T_f) \leq I(z_f, T_f)$, we can get $I(T_f, T|z_{inf}) \geq I(T_f, T|z_f)$. Consequently, $I(z_f, T) \geq I(z_{inf}, T)$ holds.

**Theorem 1.** The upper bound of error rates in downstream tasks using minimal sufficient representations is higher than that using sufficient representations.

*Proof.* For the downstream tasks, we consider the Bayes error rate (Fukunaga, 2013) to estimate the lowest achievable error of the classifier. According to the paper (Wang et al., 2022), for arbitrary representations $z$, its Bayes error rate $P_e$ satisfies that,

$$P_e \leq 1 - \exp[-H(T) + I(z, T)], \tag{5}$$

where $H(T)$ represents the entropy of variable $T$. Since $I(z_f, T) \geq I(z_{inf}, T)$, it can be concluded that the upper-bound of $P_{e,f}$ is smaller than that of $P_{e,inf}$. This indicates that ideally $z_f$ is expected to achieve better performance than $z_{inf}$ in downstream tasks.

## B MORE VISUALIZATIONS

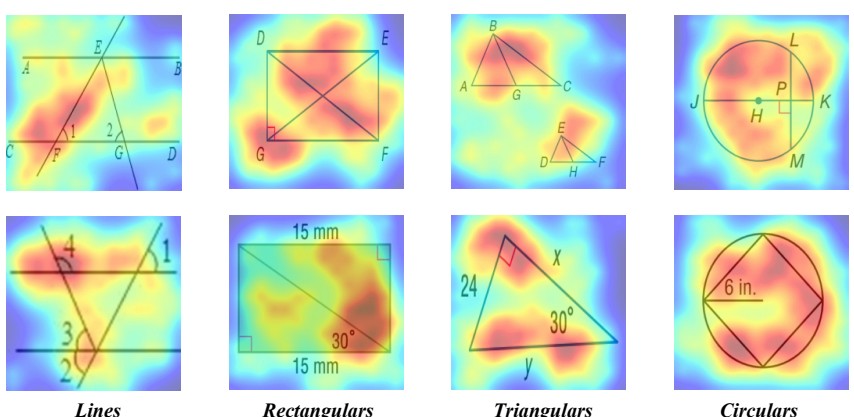

|   Lines   |   Rectangulars   |   Triangulars   |   Circulars   |

Figure 6: Attention map of GS-Former on different types of geometric diagrams.

In Fig. 6, we present attention maps of GS-Former, which show the model's attention distribution across different regions. The areas with higher brightness indicate regions considered more useful for making decisions. In contrast, darker areas are often semantically irrelevant and uninformative, which will be removed by GS-Former. This visualization highlights our model's ability to capture pivotal information across complex geometric images, such as lines, rectangles, triangles, circles, etc.

## C EXAMPLES OF FORMALIZED DIAGRAM-CAPTION PAIRS

| Image | Caption | Image | Caption |
|---|---|---|---|
| | Line A E D
Line A O C
Line B O D
Line B A
Line B C
Line C D
Line B E
Line E O | | Line B A
Line O A
Line A C
Line B O C
Line A D
Line D C
\\odot O lieson A C D B |
| | Line A O B
Line D C
Line D B
Line O C
\\odot O lieson A D C B | | Line A B
Line C D
Line E F
Line E C A
Line B D F |

Table 7: Four examples of our formalized diagram-caption pairs containing two relationships among points in geometry images.

In Table 7, we provide additional examples of descriptions that delineate the collinear and concyclic relationships in geometric images at the granularity of points. It is noteworthy that we adhered to strict grammatical and standardization criteria during the annotation process. Specifically, for collinear relationships, the term `Line` denotes the relationship, and the order of the points is listed from left to right. For concyclic relationships, the symbol `\\odot` signifies the center of the circle, `lieson` indicates the points on the circumference, and the points are listed in a clockwise order.

## D  MORE EVALUATIONS

Inspired by the Choice metric proposed by Zhang et al. (2023c), we introduce an accuracy metric for GeoX to ensure **complete fairness** when comparing with solver-free methods like G-LLaVA (Gao et al., 2023). Specifically, we observe that even if errors occur in the solving process, solver-free methods can still provide an answer (by randomly selecting one from four options), whereas our solver-based approach considers any process errors as incorrect results. To this end, in comparison with solver-free methods as shown in Tab. 8, we define GeoX's accuracy by assuming that, when the solution process encounters errors, the model's performance is equivalent to randomly selecting from four possible options. We also evaluate our method against solver-free approaches on GeoQA (Chen et al., 2021). As shown in Tab. 8, our method outperforms the current state-of-the-art solver-free methods in Top-1 accuracy.

Table 8: Comparison with solver-free geometry specialists on GeoQA. We directly report results using Top-1 accuracy.

| Methods | Base LLM | Accuracy |
|---|---|---|
| Math-LLaVA (Shi et al., 2024) | Vicuna-1.5-13B | 48.1 |
| G-LLaVA (Gao et al., 2023) | LLaMA-2-7B | 64.2 |
| MAVIS (Zhang et al., 2024) | MAmmoTH-2-7B | 66.7 |
| EAGLE (Li et al., 2024) | Vicuna-1.5-7B | 67.1 |
| GeoX (Ours) | Geo-LLM-7B | **67.4** |

## E  DATA ACQUISITION FOR GEOMETRIC CORPUS

**Data Sources.** We detail the geometric corpus collections used to train Geo-LLM, sourced from a variety of publicly available geometric datasets, including GeoQA (Chen et al., 2021), GeoQA+(Chen et al., 2021), UniGeo(Chen et al., 2022), PGDP5K (Hao et al., 2022), PGPS9K (Zhang et al., 2023c), Geometry3K (Lu et al., 2021), and G-LLaVA (Gao et al., 2023).

- **GeoQA (Chen et al., 2021)** comprises 4,998 real-world geometry problems sourced from Chinese middle school exams, each annotated with detailed solution processes and human performance metrics. The dataset is organized into three primary categories: angle, length, and other geometric calculations, and is divided into training, validation, and test sets at a ratio of 7.0:1.5:1.5.

- **Geometry3K (Lu et al., 2021)** provides 3,002 detailed geometry problems derived from high school textbooks, divided into training, validation, and test sets in a 0.7:0.1:0.2 ratio. Geometry3K expands on previous datasets (Seo et al., 2015) by including irregular quadrilaterals, polygons, and additional unknown variables and operator types. Moreover, less than 1% of Geometry3K problems can be solved without diagrams, making it more challenging.

- **GeoQA+ (Cao & Xiao, 2022)** enhances the original GeoQA (Chen et al., 2021) by adding 2,518 newly annotated geometric problems, increasing the total to 7,528 problems with 6,027 dedicated for training. This expanded dataset introduces more complex problems, including area calculations, and raises the difficulty with 27 knowledge points and an average of 2.61 solving steps per problem.

- **UniGeo (Chen et al., 2022)** introduces a comprehensive geometry dataset encompassing both calculation and proof problems, including 9,543 proving problems sourced from educational websites and 4,998 calculation problems from GeoQA (Chen et al., 2021). The proof problems are categorized into five sub-tasks (parallel, triangle, quadrangle, congruent, and similarity) with detailed reasoning and expressions. To facilitate unified problem-solving, both proofs and solutions are reformulated into sequence formats, aligning the proving steps with calculation protocols.

- **PGDP5K (Hao et al., 2022)** contains a total of 5,000 images, divided into training, validation, and test sets with a 0.7:0.1:0.2 split. It encompasses 16 geometric shapes, 5 positional relations, 16 symbol types, and 6 text types. PGDP5K provides detailed annotations, including geometric primitives, symbols, text types, and their relationships.

- **PGPS9K (Zhang et al., 2023c)** consists of 9,022 geometry problems paired with 4,000 unique diagrams, covering 30 problem types from grades 6-12 mathematics curricula. It is split into training and test sets, with 8,433 samples for training and 589 for testing. PGPS9K includes detailed annotations for diagrams and solution programs.

- **G-LLaVA (Gao et al., 2023)** is a large-scale multi-modal geometry dataset consisting of over 110k question-answer (QA) pairs, divided into an alignment dataset to provide foundational geometric knowledge and an instruction-tuning dataset to improve the model's problem-solving abilities. This dataset is created with the assistance of GPT-API (Ouyang et al., 2022) using various strategies, including equation solving, value scaling, and sentence paraphrasing.

**Data Collection and Filtering.** To meet the demands of pre-training for Geo-LLM, we build up a specialized filtering and pre-processing pipeline to construct the geometric corpus. Initially, we extract the data only from the training portions from the existing geometric datasets to prevent label leakage. Besides, we use a free Translate-API to convert Chinese data into English, ensuring language consistency. For each data entry, we remove content unrelated to geometric problems, such as annotation IDs, dates (Lu et al., 2021), and sources (Zhang et al., 2023c). Ultimately, we achieve a collection of 100 million tokens of data.

## F  ADDITIONAL DETAILS

**Prompts for MLLMs.** In Tab. 9, we provide examples of how to prompt multimodal large models to reason on geometric problems across two different evaluation modes. Each evaluation mode consists of several components: System Prompt, Diagram, Question, and optionally, Choices. The System Prompt specifies the type of problem the model is required to solve and the expected output format. The Diagram corresponds to the relevant image, while the Question and Choices are presented in the text. The key difference between the Choice and Completion modes is that Completion requires the model to provide answers directly, while Choice only involves selecting from predefined options.

**Evaluation Versions for Generalists.** In Tab. 10, we present the model / API versions utilized for the evaluation of generalists reported in Tabs. 1 to 4. These include MLLMs such as mPLUG-Owl2 (Ye et al., 2023), Qwen-VL (Bai et al., 2023), LLaVA-v1.5 (Liu et al., 2024), GPT-4V (OpenAI, 2023), and GPT-4o (OpenAI, 2024).

**Implementation Details.** After unified formal vision-language pre-training, we fine-tuned GeoX on each dataset to achieve better performance. The hyperparameters required for end-to-end visual instruction tuning are shown in Tab. 11.

| Eval Mode | Prompt |
|---|---|
| *Choice* | **System Prompt:** You are an intelligent robot expert at solving geometry problems. Please answer the Question based on the image. You should provide the reasoning process, and then you must give the correct choice in the end based on your reasoning in the following form: The answer is (A), (B), (C) or (D). 
 **Diagram**: The Diagram is `image_id.png</img>` 
 **Question:** As shown in the figure, in triangle A B C , it is known that angle A = 80.0 , angle B = 60.0 , D E parallel B C , then the size of angle C E D is (). 
 **Choices:** (A) 40.0 (B) 60.0 (C) 120.0 (D) 140.0 |
| *Completion* | **System Prompt:** You are an intelligent robot expert at solving geometry problems. Please answer the Question based on the image. You should provide the reasoning process, and then you must give the correct answer in the end based on your reasoning in the following form: e.g., The answer is [12.1]. 
 **Diagram**: The Diagram is `image_id.png</img>` 
 **Question:** Line m is the perpendicular bisector of XZ, WZ = 14.9. Find WX. |

Table 9: The prompts used for Choice and Completion modes in Multi-modal Large Language Models (MLLMs). To guide MLLMs in reasoning on geometric tasks, we adopt two evaluation modes like Zhang et al. (2023b): Choice and Completion.

| Model Name | Model / API Version |
|---|---|
| mPLUG-Owl2 (Ye et al., 2023) | mplug-owl2-llama2-7b |
| LLaVA-v1.5 (Liu et al., 2024) | llava-v1.5-13b-hf |
| Qwen-VL (Bai et al., 2023) | Qwen-VL-Chat |
| GPT-4V (OpenAI, 2023) | gpt-4-vision-preview |
| GPT-4o (OpenAI, 2024) | gpt-4o-2024-05-13 |

Table 10: Model / API versions used for evaluation across different MLLMs.

| Instruction Tuning | GeoQA | UniGeo | PGPS9K | Geometry3K |
|---|---|---|---|---|
| Training Batch Size | | 64 | | |
| Scheduler | | Cosine Annealing | | |
| Optimizer | | AdamW | | |
| Warmup Ratio | 0.05 | 0.05 | 0.05 | 0.03 |
| Epochs | 100 | 80 | 45 | 30 |
| Learning Rate | 3e-5 | 3e-5 | 6e-5 | 2e-5 |
| Evaluation Steps | 200 | 400 | 200 | 200 |

Table 11: Hyperparameters for end-to-end visual instruction tuning. We finetune these models on 4 A100 (80GB) GPUs, respectively.

## G  FURTHER DISCUSSION

**Analysis of advanced MLLMs' Ability in Formal Programs Generation.** As shown in Tab. 4, GPT-4o (OpenAI, 2024) demonstrates the highest accuracy on MathVista-GEO. In this section, we delve deeper into the few-shot learning ability of GPT-4o's in generating formalized program sequences, which are then sent to the GPS solver for solving (Chen et al., 2022). Specifically, we apply 2-shot in-context learning, providing GPT-4o (OpenAI, 2024) with two examples of formal problem-solving, along with the complete set of operation functions. Then, GPT-4o is tasked with predicting the corresponding solving program when presented with new problems and geometric images. As shown in Fig. 7, GPT-4o (OpenAI, 2024) is capable of predicting simple geometric programs, but for more complex problems, it exhibits issues such as predicting only the operation without the variable (e.g., `g_equal` in b), incorrect variables (e.g., `gougu_minus 5.0 V_1 V_2` vs `gougu_minus 5.0 V_0` in c), or wrong operations (e.g., `g_equal` vs `g_minus` in d). In contrast, GeoX can predict the correct solution in these complex and diverse cases.

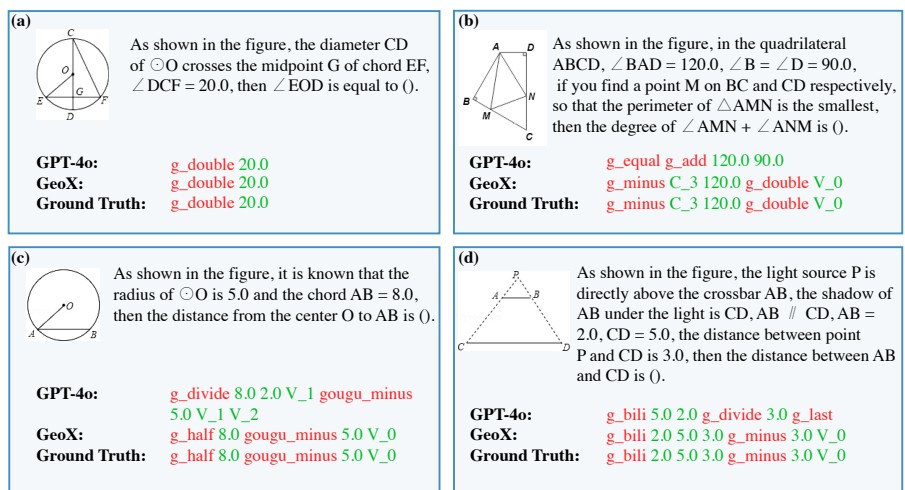

Figure 7: Comparison of GPT-4o and GeoX in predicting formalized programs for solving complex geometric problems.

