# OpenReview forum: "GeoX: Geometric Problem Solving Through Unified Formalized Vision-Language Pre-training"
_ICLR.cc/2025/Conference — ICLR 2025 Poster_

### Official Review · Reviewer_xp8T · 2024-11-02

**Soundness:** 4
**Presentation:** 3
**Contribution:** 3
**Rating:** 8
**Confidence:** 3

**Summary:**

The authors present GeoX, a multimodal large model designed specifically for tasks requiring geometric understanding and reasoning. Recognizing the distinct nature of geometric diagrams and symbols compared to natural image-text data, the authors introduce unimodal pre-training to develop a dedicated diagram encoder and symbol decoder, which enhances GeoX’s ability to process geometric images and symbolic information. Additionally, the paper proposes a novel geometry-language alignment strategy—an effective pre-training paradigm that bridges the modality gap between the unimodal geometric components. To further improve representation quality, the authors introduce a Generator-and-Sampler Transformer (GS-Former) that generates discriminative queries and filters out uninformative representations from unevenly distributed geometric signals.

**Strengths:**

The designed geometric solver significantly reduces hallucinations and incorrect results found in existing VLMs. I appreciate the design of the geometry-language alignment, which uses formalized descriptions instead of natural language captions, providing a new perspective for effectively aligning geometric and semantic features. GeoX demonstrates that formalized vision-language learning is beneficial for learning informative representations in automatic Geometry Problem Solving (GPS) tasks. GeoX can produce formalized process descriptions, enhancing both the interpretability of GPS tasks and the accuracy of the solution process.

**Weaknesses:**

Regarding the weaknesses identified in the paper, I have the following questions for the authors:

1. Computation Time of the Geometric Solver: What is the specific computation time required for the Geometric Solver? Understanding the time complexity of this component is crucial for evaluating its efficiency in practical applications.

2. Strategies to Alleviate Computational Burden: The entire procedure appears to be quite resource-intensive. Have the authors considered any strategies or methodologies to alleviate the computation cost, training time, and inference time? Exploring optimization techniques or alternative algorithms could significantly enhance the model's usability, especially in real-world scenarios where computational resources may be limited.

3. Performance Comparison with Parameters and Costs: In the performance comparison section, could the authors provide a detailed breakdown of the parameters used in their experiments, along with the associated computation costs? This information would be invaluable for readers to assess the trade-offs between performance and resource requirements when using the proposed model.

4. Ablation Study on Architecture: Is there any ablation study conducted on the architecture used for vision and language pretraining? Analyzing the contributions of different components in the model would help clarify which aspects are most beneficial for achieving the reported performance gains. This could also inform future research directions and potential improvements in model design.

**Questions:**

Since the provided examples are somewhat limited to math problems, I have concerns about the proposed approach's generalization to other geometric problems. I would consider increasing my score if the authors could validate the model on additional cases, such as geometric problems related to natural images, etc.



==========================
Thank you to the authors for the rebuttal. The results look good, and I’ve increased my score.

---

### Official Review · Reviewer_bNTG · 2024-11-03

**Soundness:** 3
**Presentation:** 3
**Contribution:** 4
**Rating:** 6
**Confidence:** 4

**Summary:**

This paper aims to align geometric representations with natural language descriptions, enhancing the model's ability to interpret and generate descriptions for complex geometric shapes and structures. This approach significantly improves the performance of existing models in accurately generating and interpreting geometric data compared to baseline methods.

**Strengths:**

1.	The introduction of geometry language alignment as a distinct framework represents a fresh perspective in the intersection of geometric representation and natural language processing.
2.	The experimental results demonstrate improvements over baseline methods, providing compelling evidence of the effectiveness of the proposed approach.

Overall, the paper presents a promising approach to geometry language alignment, and with the suggested improvements, it has the potential to make a contribution to the field.

**Weaknesses:**

The interplay between geometric representation and natural language processing is complex. The paper does not sufficiently address how these two modalities are integrated within the framework, which could be crucial for understanding the overall approach.

**Questions:**

1.	What specific features are extracted from the geometric data, and how are these features represented in the model? A detailed description of the feature extraction process would clarify this aspect.
2.	The results presented in Table 2 indicate improvements, but could the authors justify the choice of metrics used, i.e., All, Angle and Length? Are there additional metrics that could provide a more comprehensive evaluation?
3.	The paper should discuss how well the proposed model generalizes to unseen data. Are there specific scenarios or types of geometric structures where the model's performance is expected to decline?

---

### Official Review · Reviewer_z62m · 2024-11-04

**Soundness:** 3
**Presentation:** 3
**Contribution:** 3
**Rating:** 8
**Confidence:** 4

**Summary:**

This paper introduces GeoX, a multi-modal large language model (MLLMs) for the geometric problem solving (GPS) task. Previous MLLMs has limitations for understanding of both visual and symbolic information in geometry. GeoX overcomes this limitation by proposing a novel architecture, including: 1) collect a 120K pre-training dataset to train the vision encoder, symbol decoder for the diagram and language understanding; 2) Proposed Generator-And-Sampler Transformer to bridges the modality gap between geometric diagrams and formalized language; 3) With instruction tuning, this model can outperform baselines on 4 different datasets.

**Strengths:**

1. The authors found that previous vision foundation models struggled to reason geometric images effectively. They proposed an unsupervised autoencoder model to train a Geo-ViT encoder on a 120K geometry image dataset they curated. This dataset, if released, could provide valuable resources to the community.
2. The GS-Former is simple but substantially improves model performance.
3. The study also demonstrates that using formal language significantly outperforms natural language in solving geometric problems.
4. This model outperforms both generalist and specialist models by a large margin on multiple datasets.
5. The attention maps show that the model focuses on areas relevant to the question.

**Weaknesses:**

1. The numbers in Table 5 and Table 6 are inconsistent. For example, on the Geometry3K dataset, the Completion score for the model without GS-Former is listed as 33.1 in Table 5, but appears as 55.0 in Table 6. Please ensure that the settings are consistent across tables.
2. The authors argue that the CLIP encoder has limitations in processing geometric images, so they proposed a Geo-ViT encoder. However, there are no ablation studies to demonstrate the effectiveness of Geo-ViT compared to CLIP.
3. The baseline LLava score is notably low, with a Top-1 score of 9.5 on GeoQA. Did the authors finetune the LLava model on the GPS dataset, given that the original LLava fine-tuning dataset lacks GPS-related questions?

**Questions:**

1. Could the authors provide additional details for the ablation studies and clarify why the numbers differ between tables?
2. If possible, could the authors include a comparison between Geo-ViT and CLIP to highlight performance differences?
3. Have the authors finetuned Llava on the GPS dataset?

---

### Official Review · Reviewer_TRSt · 2024-11-05

**Soundness:** 2
**Presentation:** 3
**Contribution:** 2
**Rating:** 6
**Confidence:** 5

**Summary:**

This paper proposes GeoX, a multimodal large model for geometric problem solving. It consists of several training stages: training a geometry-specific ViT, a geometry-specific LLM, learning cross-modal alignment and instruction tuning. The paper specifies the necessity of using formal language for geometry figures and design a sampler to better capture visual features. Some results of GeoX are reported on geometry benchmarks.

**Strengths:**

1. The design of visual sampler is reasonable and interesting. This extracts more informative regions from the entire images.

2. Using formal language instead of natural language is important for this field.

2. The paper is easy to follow and presents good-quality figures.

**Weaknesses:**

1. The idea of training math-specific vision encoder and LLM and then aligning them has been introduced in EAGLE and MAVIS. So the training pipeline cannot be viewed as a novel contribution.

2. The evaluation benchmarks are a little narrow. How about the performance of GeoX on geometry problems in MathVista, We-Math and MathScape.

3. How does this geometry-specific model compare to SoTA multimodal large models, such as InternLM-Xcomposer, InternVL2 and QWENVL2?

**Questions:**

Please kindly see the weaknesses above.

---

### Meta-Review · Area_Chair_Ji67 · 2024-12-20

**Metareview:**

This paper introduces GeoX, a large-scale multimodal model designed specifically for tasks that require geometric understanding and reasoning. Specifically, the paper introduces unimodal pre-training to recognize the unique properties of geometric diagrams and symbols and develop specialized diagram encoders and symbol decoders. It also introduces a geometric language alignment strategy, an effective pre-training paradigm that bridges the modality gap between unimodal geometric components. Furthermore, this paper proposes a generator and sampler transformer (GS-Former) that generates discriminative queries and filters out information-poor representations from unevenly distributed geometric signals. With this approach, the GeoX model can better handle geometric data and enhance its ability to interpret and generate descriptions of complex geometric shapes and structures.

The strengths of this paper are mainly the following three points. 1) Collection of a pre-training dataset for training a vision encoder, a diagram, and a symbol decoder for language understanding. 2) Proposal of a generator and sampler transformer to bridge the modality gap between geometric diagrams and formal languages. 3) With instruction tuning, this model outperforms the baseline on four different datasets (GeoQA, UniGeo, Geometry3K, and PGPS9K).

The paper itself is well written, but it lacks a discussion of the limitations of the proposed method and a discussion of future directions.

This paper is clearly written, and all reviewers have given it positive evaluations. Based on a comprehensive judgment of the paper itself, the reviewers' comments, and the author's rebuttal, the AC considers that this paper exceeds the ICLR's acceptance threshold.

**Additional Comments On Reviewer Discussion:**

Through discussions with the reviewers, the authors made the following revisions and additions. 1) The differences between this work and previous work were clarified. 2) Comparisons were made with more SoTA multimodal large-scale models. 3) An explanation of the feature extraction process within the framework was added. 4) A discussion was held on generalizing to unknown data. 5) A performance comparison was made based on parameters and costs. 6) The proposed GeoX generalization was verified. 7) Missing information was supplemented.

---

### Decision · Program_Chairs · 2025-01-22

Accept (Poster)